# Bioactivity-Guided Isolation of Phytochemicals from *Vaccinium dunalianum* Wight and Their Antioxidant and Enzyme Inhibitory Activities

**DOI:** 10.3390/molecules26072075

**Published:** 2021-04-04

**Authors:** Tianrui Zhao, Mengxue Sun, Lingpeng Kong, Qingwang Xue, Yudan Wang, Yifen Wang, Afsar Khan, Jianxin Cao, Guiguang Cheng

**Affiliations:** 1Faculty of Agriculture and Food, Kunming University of Science and Technology, Kunming 650500, China; food363@163.com (T.Z.); smx18313941107@163.com (M.S.); 13573073577@163.com (L.K.); sdlcwyd@163.com (Y.W.); jxcao321@hotmail.com (J.C.); 2Department of Chemistry, Liaocheng University, Liaocheng 252059, China; xueqingwang1983@163.com; 3Kunming Institute of Zoology, Chinese Academy of Sciences, Kunming 650000, China; wangyifen@mail.kiz.ac.cn; 4Department of Chemistry, COMSATS University Islamabad, Abbottabad Campus, Abbottabad 22060, Pakistan; afsarhej@gmail.com

**Keywords:** *Vaccinium dunalianum*, antioxidant, enzyme inhibitory, guided isolation

## Abstract

*Vaccinium dunalianum* Wight, usually processed as a traditional folk tea beverage, is widely distributed in the southwest of China. The present study aimed to investigate the antioxidant, α-glucosidase and pancreatic lipase inhibitory activities of *V.*
*dunalianum* extract and isolate the bioactive components. In this study, the crude extract (CE) from the buds of *V. dunalianum* was prepared by the ultrasound-assisted extraction method in 70% methanol and then purified with macroporous resin D101 to obtain the purified extract (PM). Five fractions (Fr. A–E) were further obtained by MPLC column (RP-C18). Bioactivity assays revealed that Fr. B with 40% methanol and Fr. D with 80% methanol had better antioxidant with 0.48 ± 0.03 and 0.62 ± 0.01 nM Trolox equivalent (TE)/mg extract for DPPH, 0.87 ± 0.02 and 1.58 ± 0.02 nM TE/mg extract for FRAP, 14.42 ± 0.41 and 19.25 ± 0.23 nM TE/mg extract for ABTS, and enzyme inhibitory effects with IC_50_ values of 95.21 ± 2.21 and 74.55 ± 3.85 for *α*-glucosidase, and 142.53 ± 11.45 and 128.76 ± 13.85 µg/mL for pancreatic lipase. Multivariate analysis indicated that the TPC and TFC were positively related to the antioxidant activities. Further phytochemical purification led to the isolation of ten compounds (1–10). 6-*O*-Caffeoylarbutin (7) showed significant inhibitory effects on *α*-glucosidase and pancreatic lipase enzymes with values of 38.38 ± 1.84 and 97.56 ± 7.53 µg/mL, and had the highest antioxidant capacity compared to the other compounds.

## 1. Introduction

Discovering bioactive compounds from natural sources is a growing research interest nowadays. To some extent, consumption of phenolics presented in food resources may reduce the risk of some chronic diseases such as abnormal cholesterol metabolism, type 2 diabetes mellitus, and hypertension [1,2]. Phenolic compounds are important natural plant secondary metabolites. They are usually classified as phenolic acids, flavonoids, tannins, lignans, and stilbene derivatives due to the structural skeletons [3]. These compounds exhibit a wide range of biological properties such as antioxidant, anti-diabetic, anti-inflammatory, anticancer, antimicrobial, and cytoprotective activities [4,5]. Attention has been focused on the phenolics, especially flavonoids, because they are able to inhibit lipid peroxidation [6].

The balance between the generation and elimination of free radicals and reactive oxygen species (ROS) plays an important role in human health. Oxidative stress occurs through a disturbance of the antioxidant balance. Oxidative stress is closely related to human diseases such as aging, atherosclerosis, diabetes mellitus, and rheumatoid arthritis. It is reported that oxidative stress could decrease the expression of insulin gene, insulin secretion, and even damage the islet [7]. In addition, *α*-glucosidase is a key enzyme responsible for the digestion of carbohydrates and subsequent glucose absorption [8,9]. Thus, there is an urgent need to search for effective antioxidant ingredients and α-glucosidase inhibitors in the management of type 2 diabetes.

*Vaccinium dunalianum* Wight, a traditional medicinal and edible plant, is mainly distributed in the southwest of China such as Sichuan, Guizhou, and Yunnan Provinces. The Yi people in its growing area of Yunnan Province, China use the leaves and leaf buds as a tea and the whole plant is used medicinally to treat rheumatoid arthritis pain. In some minority areas, herbal plants are usually processed as a substitute of tea and are used as a traditional remedy for alleviating hypertension, diabetes, atherosclerosis, and maintaining healthy kidneys as well as relieving coughs [10,11]. Its buds are traditionally consumed as a folk health tea beverage, known as “Quezui tea”, with multiple health benefits [12]. It is also used as a traditional Chinese medicine in the treatment of diabetes, hyperlipemia, and articular rheumatism [13,14,15].

Modern investigations have reported that *Vaccinium* plants have a range of antioxidants such as flavonoids, anthocyanins, proanthocyanidins, and so on, which could prevent oxidative damage caused by ROS [16,17]. Previous studies on *V. dunalianum* led to the isolation of caffeic acid derivatives, arbutin derivatives, and flavonoids [11,13,18]. However, these studies have mainly been focused on the phytochemicals with novel structures of arbutin derivatives. Anti-diabetic and hypolipidemic activity evaluations of the extract and pure compound from *V. dunalianum* are limited.

In order to investigate the bioactive ingredients of *V. dunalianum*, its methanol extract was prepared, purified, and fractioned by D101 macroporous resin column chromatography and MPLC. Under the antioxidant and *α*-glucosidase, and pancreatic lipase inhibitory assays, bioactivity-guided isolation was performed to find out the potential bioactive phytochemicals corresponding for its tradition applications in human health.

## 2. Results and Discussion

### 2.1. Bioguided Isolation and Structural Characterization of Isolated Compounds

The crude extract of *V. dunalianum* buds was purified to obtain a purified extract with phenolic enrichment through D-101 macroporous resin column chromatography. The PM had better inhibitory effects on *α*-glucosidase and pancreatic lipase, and antioxidant capacities than those of CE (Table 1). Furthermore, the purified material (PM) was further fractionated into five fractions by macroporous resin column chromatography including Fr. A eluted with 20% methanol, Fr. B with 40% methanol, Fr. C with 60% methanol, Fr. D with 80% methanol, and Fr. E with 100% methanol, successively. Fr. B and Fr. D showed better antioxidant and inhibitory activity against pancreatic lipase and *α*-glycosidase compared to other fractions. Although Fr. C had a higher TPC concentration (36.11 ± 0.12 mg GAE/mg extract), we found that this fraction was rich in 6′-*O*-caffeoylarbutin (**7**). According to the bioassay-guided fractionation and isolation, compounds **1**–**7** were isolated from the active fraction B, while compounds **8**–**10** were obtained from fraction D. As shown in Figure 1, the isolates include three arbutin derivatives (**1**, **7**, and **8**), four caffeic acid derivatives (**2**, **3**, **4**, and **6**), one phenolic acid (**10**), and two flavonoid glycosides (**5** and **9**). In addition, quercetin-3-*O*-arabinofuranoside (**5**) and caffeic acid ethyl ester (**6**) were isolated for the first time from this plant.

### 2.2. Quantification of the Isolated Compounds in the Extracts and Fractions

The chemical constituents in the CE, PM, and fractions were analyzed by UHPLC-ESI-HRMS/MS operated in a negative mode. The total ion chromatograms of various extracts are presented in Figure 2. It is clearly revealed that the chemical constituents were significantly different in polarity fractions. It is also shown that the chemical components were effectively separated into different fractions via MPLC. The isolated compounds were used as standards for the identification and quantification of chemical components in the CE, PE, and fractions. The peaks in the UHPLC-ESI-HRMS/MS chromatographic profiles were readily assigned as follows: arbutin (t_R_ = 1.49 min, peak 1), 6-*O*-*E*-caffeoyl-d-glucopyranose (t_R_ = 3.18 min, peak 2), caffeic acid (t_R_ = 5.08 min, peak 3), caffeic acid methyl ester (t_R_ = 5.65 min, peak 4), quercetin-3-*O*-arabinofuranoside (t_R_ = 6.58 min, peak 5), caffeic acid ethyl ester (t_R_ = 7.37 min, peak 6), 6′-*O*-caffeoylarbutin (t_R_ = 8.43 min, peak 7), robustaside A (t_R_ = 10.20 min, peak 8), kaempferol-3-*O*-*β*-d-glucopyranoside (t_R_ = 11.25 min, peak 9), and *p*-hydroxy benzoic acid (t_R_ = 11.69 min, peak 10), respectively (Table 2). The MS/MS data are also presented in Table 1.

The content of each compound in different extracts is described and expressed as µg/g extract in Table 2. Quantitative results indicated that 6′-*O*-caffeoylarbutin (**7**) was the most abundant component in CE and PM, which was consistent with the reported data [13]. After MPLC fractionation, 6′-*O*-caffeoylarbutin mainly existed in Fr. C, followed by Fr. D, B, A, and Fr. E. Arbutin was completely found in the Fr. A with 48,580.19 ± 336.91 µg/g extract, which was 1.41 times of that in the PM. Furthermore, the most abundant compound in Fr. D was robustaside A at 298,710.50 ± 3023.29 µg/g extract, which was approximately 14.40 and 4.77 times higher than those of the PM and CE, respectively.

The chemical constituents of the fractions from PM were significantly influenced by the eluting solvent. Considering the solubility of phenolic compounds depending on their structural guided polarity, the chemical compounds in *V. dunalianum* may possess relatively higher polarities, which are compatible with that of the elution solvent comprising of a lower methanol/water ratio. When compared to the CE, the higher contents of 10 compounds were observed, which might be related to the removal of water-soluble salts, proteins, and sugars [19].

### 2.3. Total Phenolic Contents (TPC) and Total Flavonoid Contents (TFC)

In the present study, the TPC and TFC of the crude extract and fractions with different polarity from *V. dunalianum* buds were analyzed to obtain a bioactive enrichment fraction. The TPC is expressed as gallic acid (GAE) equivalent (mg GAE/g extract) and TFC is expressed as rutin (RE) equivalent (mg RE/g extract). The TPC and TFC of each extract are presented in Table 3. The results revealed that PM showed higher TPC and TFC than those of CE. The value of TPC was 36.16 ± 0.19 µg GAE/mg extract, which was 2.96 times of that in the CE. The value of TFC was 14.39 ± 0.19 µg RE/mg extract, which was approximately 2.80 times that in the CE. In terms of fractions, Fr. D had the highest values of TPC (56.15 ± 0.39 mg GAE/g extract) and TFC (20.31 ± 0.18 µg RE/g extract), followed by Fr. C with TPC values of 36.11 ± 0.32 mg GAE/g extract and TFC of 5.64 ± 0.29 µg RE/mg extract. In Fr. C, 6′-*O*-caffeoylarbutin, a phenolic compound, was the most abundant compound. Fr. B had moderate values of TPC (30.93 ± 0.23 mg GAE/g extract) and TFC (9.42 ± 0.19 µg RE/mg extract). It was obviously revealed that the solvent with different polarities had a significant effect on polyphenol and flavonoid content in the fractions.

### 2.4. The α-Glucosidase and Pancreatic Lipase Inhibitory Activities

The inhibitory effects of CE, PM, fractions, and isolates are summarized in Table 1. The PM had better α-glucosidase and pancreatic lipase inhibitory activities with IC_50_ values of 109.75 ± 1.34 and 165.43 ± 10.35 µg/mL than those of the crude extract (IC_50_ = 198.56 ± 9.34 and 214.32 ± 17.53 µg/mL, respectively). It is reported that macroporous resin column chromatography could effectively enrich the polyphenols in the purified extract and exhibit better anti-oxidative and cytoprotective activities [20]. After macroporous resin chromatography, the sample showed more significant bioactivity in antioxidant, enriched total phenolic, and flavonoids. After macroporous resin chromatography, TPC and TFC were completely enriched in the PM. Among the five fractions, Fr. B and Fr. D exerted the most potent inhibition of *α*-glucosidase (95.21 ± 2.21 and 74.55 ± 3.85 µg/mL, respectively) and pancreatic lipase (142.53 ± 11.45 and 128.76 ± 13.85 µg/mL, respectively) than the CE and PM. Fr. A and Fr. C showed relatively weak α-glucosidase and pancreatic lipase inhibitions, whereas Fr. E exhibited the lowest inhibitory activity against α-glucosidase and pancreatic lipase (303.03 ± 7.27 and 589.65 ± 10.98 µg/mL, respectively). The results showed that the greater the TPC and/or TFC in different fractions from *V. dunalianum* buds, in general, the greater the values of inhibitory activity on enzymes.

In addition, the main bioactive compounds corresponding to enzyme inhibitory activities should be explored. As shown in Table 1, all of the isolated compounds (**1**–**10**) showed significant inhibitory effect against the α-glucosidase enzyme. 6′-*O*-Caffeoylarbutin (**7**) and 6-*O*-*E*-caffeoyl-d-glucopyranose (**2**) exhibited the strongest α-glucosidase inhibition with IC_50_ values of 38.38 ± 1.84 and 42.43 ± 2.87 µg/mL. Arbutin (**1**), quercetin-3-*O*-arabinofuranoside (**5**), caffeic acid ethyl ester (**6**), robustaside A (**8**), kaempferol-3-*O*-*β*-d-glucopyranoside (**9**), and *p*-hydroxy benzoic acid (**10**) had similar and relatively weak inhibitory activity against α-glucosidase, while caffeic acid (**3**) showed the lowest inhibitory activity (IC_50_ = 70.68 ± 0.73 µg/mL). For pancreatic lipase inhibitory effects, 6′-*O*-caffeoylarbutin (**7**) and 6-*O*-*E*-caffeoyl-d-glucopyranose (**2**) had the most remarkable inhibitory activity (97.56 ± 7.53 and 103.86 ± 3.01 µg/mL, respectively). Therefore, **7** and **2** may be the most active compounds responsible for in vitro inhibitory activity against α-glucosidase and pancreatic lipase enzymes. Fr. C possessed the maximum content of compound **7**, but was not the most bioactive fraction. This result may be related to the multiple-compound synergy effects in a combination of **7** with other compounds [21]. In addition, caffeic acid methyl ester (**4**) and caffeic acid ethyl ester (**6**) have been reported to have good *ɑ*-glucosidase inhibitory activities, which are responsible for its traditional use in the treatment of diabetes [22]. The significance of our study is that α-glucosidase activity is associated with many diseases such as diabetes mellitus, diabetic peripheral vascular disease, diabetic cardio-cerebrovascular complications, glucose and lipid metabolism disorders, atherosclerosis, coronary heart disease and other cardio-cerebrovascular diseases, obesity, and fatty liver disease [23].

### 2.5. Antioxidant Activities

In this study, three commonly antioxidant methods (DPPH, FRAP, and ABTS) were applied to evaluate the antioxidant potential of CE, PM, fractions, and compounds. The DPPH and ABTS radical scavenging capacity and FRAP value were expressed as nmol Trolox equivalent (TE)/g extract. The antioxidant results are presented in Table 1. When compared to the CE, PM exerted better antioxidant capacity for DPPH (0.40 ± 0.02 nmol TE/mg extract), FRAP (0.91 ± 0.04 nmol TE/mg extract), and ABTS (12.87 ± 0.13 nmol TE/mg extract). After MPLC purification, Fr. D had the highest antioxidant capacity for DPPH (0.62 ± 0.01 nmol TE/mg extract), FRAP (1.58 ± 0.02 nmol TE/mg extract), and ABTS (19.25 ± 0.23 nmol TE/mg extract), followed by Fr. B.

Almost all the isolated compounds showed good antioxidant activity (Table 2). Among these compounds, 6′-*O*-caffeoylarbutin (**7**) had the best antioxidant activity in DPPH (1.21 ± 0.43 nmol TE/g extract), FRAP (0.87 ± 0.01 nmol TE/g extract), and ABTS (16.96 ± 0.95 nmol TE/g extract), followed by 6-*O*-*E*-caffeoyl-d-glucopyranose (**2**). The results were similar to those of phenolic substances in rabbit eye blueberry (*Vaccinium ashei*) [24]. The values of TPC and TFC were highly correlated with the DPPH values (r = 0.9114 and 0.7484), and strongly with FRAP (r = 0.9076 and 0.9205) and ABTS (r = 0.9311 and 0.8058). These results revealed that the TPC or TFC content are responsible for antioxidant capacity, which were consistent with the results observed in a previous study [25]. The content of robustaside A (**8**), kaempferol-3-*O*-*β*-d-glucopyranoside (**9**), and 4-hydroxy benzoic acid (**10**) highly correlated with the values of antioxidants (r = 0.8948, 0.8469, and 0.8691, respectively, *p* < 0.05). It has been reported that the antioxidant capacity of different fractions is positively associated with phenolics [26]. In addition, we found the content of 6′-*O*-caffeoylarbutin, which had a moderate correlation with the values of α-glucosidase and pancreatic lipase inhibitory activities (r = −0.7104 and −0.6919, respectively, *p* < 0.05). There are also studies that have shown that caffeic acid (**3**) [27], quercetin-3-*O*-arabinofuranoside (**5**) [28], and kaempferol-3-*O-β-*d-glucopyranoside (**9**) have a positive effect on antioxidant activity [29].

### 2.6. Multivariate Analysis (PCA)

The principal component analysis (PCA) extracted from the data of Table 1 explained 96.91% of total variation, where PC1 accounts for 90.73% of the variance and PC2 for 6.18% (Figure 3). Statistical study indicated that there were some correlations between the obtained results. The upper right quadrant in the negative side of PC1 includes Fr. B and Fr. C. Fr. C was characterized by high TPC, whereas Fr. B was mostly characterized by high antioxidant activity (DPPH, ABTS, and FRAP) and high total phenols and flavonoids. The PM and Fr. D in the lower right quadrant represent the DPPH, ABTS, and FRAP capacity. It also had a high content of total phenols and flavonoids. Associated with the antioxidant capacity values, Fr. A and E had low TPC and TFC concentrations and were located along the axis of PC2, having negative scores. Furthermore, the TPC and/or TFC were closely associated with antioxidant activities. Phenolics and flavonoids may be the main antioxidant compounds [30]. This finding is consistent with the chemical constituent in each extract and the antioxidant activities of individuals.

## 3. Materials and Methods

### 3.1. General

LC/MS grade methanol and acetonitrile were purchased from Merck (Darmstadt, Germany). Ultrapure water was deionized using a Milli-Q purification system (Millipore, Bedford, MA, USA). 2,2-Diphenyl-1-picryhydrazyl (DPPH), 2,2’-azino-bis (3-ethylbenzothiazoline-6-sulfonic acid) (ABTS), Trolox, Folin-Ciocalteu reagent, α-glucosidase, pancreatic lipase, 3,5-dinitrosalicylic acid (DNS), p-nitrophenyl α-d-glucopyranoside (PNPG), acarbose (95%), and orlista (purity ≥ 97.0%) were purchased from Sigma-Aldrich (Shanghai, China). Silica gel (200–300 mesh, Qingdao Marine Chemical Ltd., Qingdao, China) Chromatores C18 (20–45 µm, Fuji Silysia Chemical Ltd., Tokyo, Japan), Sephadex LH-20 (Pharmacia Fine Chemical Co. Ltd., Uppsala, Sweden) were used for column chromatography. MPLC was carried out on a Büchi pump system coupled with C18 silica gel-packed glass columns (15 × 230 and 26 × 460 mm). HPLC was performed on an Agilent 1260 liquid chromatograph (Agilent Technologies Co. Ltd., Palo Alto, CA, USA) coupled with Zorbax SB-C18 columns (9.4 × 250 and 21.2 × 250 mm). 1D NMR spectra were recorded on a Bruker DRX-500 MHz spectrometer (Bruker Co., Karlsruhe, Germany) with TMS as an internal standard. HRESIMS data were recorded on a Q-Exactive Focus mass spectrometer (Thermo Fisher Scientific, Bremen, Germany). All the other chemicals and reagents (analytical grade) were purchased from Sinopharm Chemical Reagent Co. Ltd. (Shanghai, China).

### 3.2. Plant Material

The *V. dunalianum* buds were collected from Chuxiong Yi Autonomous Prefecture, Yunnan Province, China on 9 May 2019, and identified by Prof. Jianxin Cao (one of the authors). A voucher specimen (no. Cheng2019050901) has been deposited at the Faculty of Agriculture and Food, Kunming University of Science and Technology, China.

### 3.3. Extraction and Fractionation

The collected buds of *V. dunalianum* were dried in a shade room with good ventilation and without direct exposure to sunlight at 20 ± 5 °C until constant weight. The dried sample (4.0 kg) was powdered and passed by a 100-mesh sieve, then extracted with 70% methanol–water (8.0 L) by ultrasound assisted extraction (UAE) (400 W) three times (30 min each time). After centrifugation at 4000 g for 10 min, the supernatant was collected and evaporated in vacuo to yield a crude methanol extract (CE, 747 g). The CE was dissolved in water and subjected to a macroporous resin (D101) column chromatography to obtain the purified extract (PM, 531 g). Furthermore, the PM was fractioned through macroporous resin (D101) column chromatography, eluting with the methanol–water system (20, 40, 60, 80, and 100%). The extraction and isolation are schematically outlined in Figure 4. The water eluent was discarded, and the methanol–water elution fractions were collected and dried to obtain five fractions (Fr. A–E, 70 g, 56 g, 65 g, 24 g, and 101 g) (Figure 4).

### 3.4. Bio-Guided Isolation of Active Constituents

Fractions B (56 g) and D (24 g) exhibited significant antioxidant activity and were selected for further isolation (Figure 4). Fr. B was chromatographed on a silica gel column (CHCl_3_/MeOH, 30:1–1:1) to afford compound 7 (23 g) and a subfraction Fr. B-1. The subfraction Fr. B-1 was purified on a silica gel column with a solvent mixture of CHCl_3_/MeOH, (30:1, 20:1, 10:1, 5:1, and 3:1) and further purified by a HPLC preparation column (MeOH/H_2_O; 1:1) to yield compounds **1** (8 g), **2** (26 mg), **3** (1.89 g), **4** (2.45 g), **5** (845 mg), and **6** (170 mg). Fr. D was subjected to MPLC with a MeOH/H_2_O gradient (20:80–60:40) to yield Fr. D1–D5. Fr. D3 (7 g) was separated through preparative HPLC with a gradient of CH_3_CN/H_2_O (25:75–45:55) to afford compound **8** (1.25 g). Fr. D5 (5 g) was chromatographed by MPLC with CH_3_OH/H_2_O (from 20:80 to 70:30) and purified by preparative HPLC with a gradient of CH_3_OH/H_2_O (30:70–60:40) to yield compounds **9** (16 mg) and **10** (15 mg).

### 3.5. Structural Characterization of Compounds

The structures of the isolates were identified by interpreting NMR (^1^H and ^13^C NMR) spectra and ESI-MS analysis. The chemical structures of the compounds are shown in Figure 1.

Arbutin (**1**), white amorphous powder, C_12_H_16_O_7_, HR-ESI-MS *m/z*: 271.0821 [M−H]ˉ (calcd for C_12_H_15_O_7_, 271.0823); ^1^H NMR (500 MHz, CD_3_OD) *δ*_H_: 6.97 (2H, d, *J* = 8.9 Hz, H-2/H-6), 6.69 (2H, d, *J* = 8.9, H-3/H-5), 4.73 (1H, d, *J* = 7.2, H-1′); ^13^C NMR (CD_3_OD, 125 MHz) δC: 62.4 (C-6′), 71.3 (C-4′), 74.9 (C-2′), 78.0 (C-3′/C-5′), 103.4 (C-1′), 116.5 (C-3/C-5), 119.4 (C-2/C-6), 152.5 (C-1), 153.8 (C-4). The ^1^H and ^13^C NMR spectroscopic data were in accordance with those of arbutin [31].

6-*O*-*E*-Caffeoyl-d-glucopyranose (**2**), white amorphous powder, C_15_H_18_O_9_, HR-ESI-MS *m/z*: 341.0877 [M−H]ˉ (calcd for C_15_H_17_O_9_, 341.0878); ^1^H NMR (500 MHz, CD_3_OD) *δ*_H_: 7.59 (1H, d, *J* = 15.8 Hz, H-7), 7.04 (1H, d, *J* = 1.9 Hz, H-2), 6.78 (1H, d, *J* = 8.1 Hz, H-5), 6.95 (1H, dd, *J* = 8.1, 1.9 Hz, H-6), 6.30 (1H, d. *J* = 15.8 Hz, H-8); ^13^C NMR (125 MHz, CD_3_OD) *δ*_C_: 64.9 (C-6′), 70.9 (C-5′), 72.1 (C-4′), 74.0 (C-2′), 75.6 (C-3′), 94.1 (C-1′), 114.8 (C-8), 115.1 (C-2), 116.2 (C-5), 122.7 (C-6), 127.2 (C-1), 149.2 (C-7), 169.2 (C-9). The ^1^H and ^13^C NMR spectroscopic data were in accordance with those of 6-*O*-trans-caffeoyl-d-glucopyranose [32].

Caffeic acid (**3**), white crystal, C_9_H_8_O_4_, HR-ESI-MS *m/z*: 179.0341 [M−H]^−^ (calcd for C_9_H_7_O_4_, 179.0350); ^1^H NMR (500 MHz, CD_3_OD) *δ*_H_: 7.49 (1H, d, *J* = 16.0 Hz, H-7), 7.07 (1H, d, *J* = 2.0 Hz, H-2), 6.99 (1H, dd, *J* = 8.6, 2.0 Hz, H-6), 6.77 (1H, d, *J* = 8.6 Hz, H-5), 6.28 (1H, d, *J* = 16.0 Hz, H-8); ^13^C NMR (125 MHz, CD_3_OD) *δ*_C_: 113.6 (C-2), 114.7 (C-5), 115.7 (C-8), 121.4 (C-6), 125.4 (C-1), 145.2 (C-7), 145.6 (C-3), 148.5 (C-4), 166.9 (C-9). The ^1^H and ^13^C NMR spectroscopic data were in accordance with those of caffeic acid [33].

Caffeic acid methyl ester (**4**), yellow amorphous powder, C_10_H_10_O_4_, HR-ESI-MS *m/z*: 193.0491 [M−H]^−^ (calcd for C_10_H_9_O_4_, 193.0506); ^1^H NMR (500 MHz, CD_3_OD) δH: 7.49 (1H, d, *J* = 15.6 Hz, H-7), 7.07 (1H, d, *J* = 2.0 Hz, H-2), 6.99 (1H, d, *J* = 8.4, 2.0 Hz, H-6), 6.77 (1H, d, *J* = 8.4 Hz, H-5), 6.28 (1H, d, *J* = 15.6 Hz, H-8), 3.68 (3H, -OCH_3_);. ^13^C NMR (125 MHz, CD_3_OD) *δ*_C_: 51.2 (-OCH_3_), 113.6 (C-2), 125.4 (C-1), 145.6 (C-3), 148.5 (C-4), 114.7 (C-5), 121.4 (C-6), 145.2 (C-7), 115.7 (C-8), 166.9 (C-9). ^1^H and ^13^C NMR spectroscopic data were in accordance with those of caffeic acid methyl ester [34].

Quercetin-3-*O*-arabinofuranoside (**5**), white amorphous powder, C_20_H_18_O_11_, HR-ESI-MS *m/z*: 433.0847 [M−H]ˉ (calcd for C_20_H_17_O_11_, 433.0776); ^1^H NMR (500 MHz, CD_3_OD)*δ*_H_: 7.72 (1H, d, *J* = 2.1 Hz, H-2′), 7.57 (1H, d, *J* = 8.2, 2.1 Hz, H-6′), 6.99 (1H, d, *J* = 8.2 Hz, H-5′), 6.52 (1H, d, *J* = 2.2 Hz, H-8), 6.28 (1H, d, *J* = 2.2, H-6), 5.49 (1H, d, *J* = 5.0 Hz, H-1′′); ^13^C NMR (125 MHz, CD_3_OD) *δ*_C_: 48.1 (C-5′′), 61.1 (C-4″), 77.3 (C-2′′), 81.9 (C-3″), 93.4 (C-8), 98.5 (C-6), 104.2 (C-1″), 108.0 (C-10), 115.0 (C-2′), 115.4 (C-5′), 121.5 (C-1′), 121.7 (C-6′), 133.5 (C-3), 145.0 (C-3′), 148.5 (C-4′), 157.2 (C-2), 157.9 (C-9), 161.7 (C-5), 164.8 (C-7), 178.6 (C-4). The ^1^H and ^13^C NMR spectroscopic data were in accordance with those of quercetin-3-*O*-arabinofuranoside [33].

Caffeic acid ethyl ester (**6**), yellow amorphous powder, C_11_H_12_O_4_, HR-ESI-MS *m/z*: 207.0611 [M−H]^−^ (calcd for C_11_H_11_O_4_, 207.0663); ^1^H NMR (500 MHz, CD_3_OD) *δ*_H_: 7.47 (1H, d, *J* = 15.9 Hz, H-7), 7.05 (1H, *J* = 2.0 Hz, H-2), 6.99 (1H, *J* = 8.2, 2.0 Hz, H-6), 6.77 (1H, d, *J* = 8.2 Hz, H-5), 6.28 (1H, d, *J* = 15.9 Hz, H-8), 1.17 (3H, t, *J* = 7.0 Hz, H-2′), 4.08 (2H, q, *J* = 7.0 Hz, H-1′); ^13^C NMR (125 MHz, CD_3_OD) *δ*_C_: 14.6 (C-2′), 58.9 (C-1′), 113.6 (C-2), 114.7 (C-5), 115.7 (C-8), 121.4 (C-6), 125.4 (C-1), 145.2 (C-7), 145.6 (C-3), 148.5 (C-4), 166.9 (C-9). ^1^H and ^13^C NMR spectroscopic data were in accordance with those of caffeic acid ethyl ester [35].

6′-*O*-Caffeoylarbutin (**7**), white amorphous powder, C_21_H_22_O_10_, HR-ESI-MS *m/z*: 433.1137 [M−H]ˉ (calcd for C_21_H_21_O_10_, 433.1140); ^1^H NMR (500 MHz, CD_3_OD) *δ*_H_: 7.57 (1H, d, *J* = 15.6 Hz, H-7″), 7.04 (1H, d, *J* = 2.0 Hz, H-2″), 6.95 (1H, dd, *J* = 8.4, 2.0 Hz, H-6″), 6.78 (1H, d, *J* = 8.4 Hz, H-5″), 6.66 (2H, d, *J* = 8.4 Hz, H-3/H-5), 6.63 (2H, d, *J* = 8.4 Hz, H-2/H-6), 6.29 (1H, d, *J* = 15.6 Hz, H-8″), 4.74 (1H, d, *J* = 7.3 Hz, H-1′); ^13^C NMR (125 MHz, CD_3_OD) *δ*_C_: 64.6 (C-6’), 71.8 (C-4′), 74.9 (C-2′), 75.5 (C-5′), 77.9 (C-3′), 103.7 (C-1′), 114.8 (C-8′′), 115.1 (C-2′′), 116.6 (C-5′′), 116.7 (C-3/C-5), 119.6 (C-2/C-6), 123.2 (C-6′′), 127.7 (C-1′′), 146.8 (C-4′′), 147.2 (C-7′′), 149.7 (C-3′′), 152.3 (C-4), 153.9 (C-1), 169.0 (C-9′′). The ^1^H and ^13^C NMR spectroscopic data were in accordance with those of 6′-*O-*caffeoylarbutin [36].

Robustaside A (**8**), white amorphous powder, C_21_H_22_O_9_, HR-ESI-MS *m/z*: 417.1224 [M−H]^−^ (calcd for C_21_H_21_O_9_, 417.1191); ^1^H NMR (500 MHz, CD_3_OD) *δ*_H_: 7.63 (1H, d, *J* = 15.9 Hz, H-7″), 7.45 (2H, d, *J* = 8.6 Hz, H-2″/H-6″), 6.94 (2H, d, *J* = 8.2 Hz, H-2/H-6), 6.81 (2H, d, *J* = 8.6 Hz, H-3″/H-5″), 6.65 (2H, d, *J* = 8.2 Hz, H-3/H-5), 6.34 (1H, d, *J* = 15.9 Hz, H-8″), 4.73 (1H, d, *J* = 7.2 Hz, H-1′), 4.53 (1H, dd, *J* = 11.9, 2.0 Hz, H-6a′), 4.34 (1H, dd, *J* = 11.9, 6.7 Hz, H-6b′), 3.64 (1H, t, *J* =7.0 Hz, H-5′), 3.44 (3H, m); ^13^C NMR (125 MHz, CD_3_OD) *δ*_C_: 64.7 (C-6′), 71.7 (C-4′), 74.9 (C-2′), 75.4 (C-5′), 77.9 (C-3′), 103.7 (C-1′), 114.8 (C-8″), 116.6 (C-3/C-5), 116.9 (C-3″/5″), 119.6 (C-2/C-6), 127.1 (C-1″), 131.2 (C-2″/C-6″), 146.8 (C-7″), 152.3 (C-4), 153.9 (C-1), 161.3 (C-4″), 169.0 (C-9″). The ^1^H and ^13^C NMR spectroscopic data were in accordance with those of robustaside A [37].

Kaempferol-3-*O*-β-d-glucopyranoside (**9**), white amorphous powder, C_2_1H_20_O_11_, HR-ESI-MS *m/z*: 447.0996 [M−H]^−^ (calcd for C_21_H_19_O_11_, 447.0933); ^1^H NMR (500 MHz, CD_3_OD) *δ*_H_: 8.04 (2H, d, *J* = 8.6 Hz, H-2′/H-6′), 6.88 (2H, d, *J* = 8.6 Hz, H-3′/H-5′), 6.44 (1H, d, *J* = 2.0 Hz, H-8), 6.21 (1H, d, *J* = 2.0 Hz, H-6), 5. 41 (1H, d, *J* = 7.2 Hz, H-1″); ^13^C NMR (125 MHz, CD_3_OD) *δ*_C_: 61.2 (C-6′′), 69.9 (C-4′′), 74.3 (C-2′′), 76.6 (C-3′′), 77.0 (C-5′′), 93.3 (C-8), 98.5 (C-6), 101.1 (C-1′′), 102.6 (C-10), 114.7 (C-3′/C-6′), 120.9 (C-1′), 130.8 (C-2′/C-6′), 133.5 (C-3), 156.3 (C-9), 156.5 (C-2), 159.1 (C-4′), 161.4 (C-5), 164.2 (C-7), 177.7 (C-4). The ^1^H and ^13^C NMR spectroscopic data were in accordance with those of kaempferol-3-*O*-*β*-d-glucopyranoside [38].

*p*-Hydroxybenzoic acid (**10**), white amorphous powder, C_7_H_6_O_3_, HR-ESI-MS *m/z*: 137.0233 [M−H]^−^ (calcd for C_7_H_5_O_3_, 137.0244); ^1^H NMR (500 MHz, CD_3_OD) *δ*_H_: 7.87 (2H, d, *J* = 8.5 Hz, H-3/H-5), 6.81 (2H, d, *J* = 8.5 Hz, H-2/H-6). ^13^C NMR (125 MHz, CD_3_OD) *δ*_C_: 116.0 (C-3/C-5), 122.9 (C-1), 132.9 (C-2/C-6), 163.2 (C-4), 170.4 (COOH). The ^1^H and ^13^C NMR spectroscopic data were in accordance with those of *p*-hydroxybenzoic acid [38].

### 3.6. Enzyme Inhibitory Activity

#### 3.6.1. The Inhibitory Activity on α-Glucosidase

The α-glucosidase inhibitory assay was performed as previously described with slight modifications [39]. A 50 µL sample (20, 50, 100, 200, 400, 600 µg/mL) solution was added to the mixture of the α-glucosidase solution, phosphate buffer (100 µL, pH 6.8), and 50 µL of 10 mM PNPG in a 96-well microplate. Subsequently, the reaction mixture was incubated for 15 min at 37 °C. Then, sodium carbonate (50 µL, 0.2 M) was added to terminate the reaction. A blank was prepared for each sample without enzyme solution to correct the background absorbance (control). The absorbance (A) of the reaction mixture was measured at 405 nm by using a SpectraMax M5 microplate reader (Molecular Devices, Sunnyvale, CA, USA). The inhibitory percentage was determined as follows:Inhibition (%) = [(Acontrol − Asample)/Acontrol] × 100(1)

#### 3.6.2. The Inhibitory Activity on Pancreatic Lipase

The pancreatic lipase inhibitory assay was conducted as previously described with some modifications [40]. First, the porcine pancreatic lipase solution was centrifuged to collect the supernatant enzyme at 4000 g for 5 min. The substrate solution was prepared with p-nitrophenyl laurate dissolved in 5 mM sodium acetate (1:100 *m*/*v*) and Triton X-100 (1:100 *v*/*v*). The substrate solution was placed in boiling water for 2 min to fully dissolve, and then cooled down to room temperature for the enzymatic assay. The sample was dissolved in DMSO (<5% in the final reaction reagents) and diluted with 100 mM Tris-buffer (pH 8.2). Samples of 50, 100, 200, 400, 600, and 800 µg/mL were prepared. Orlistat was used as a positive control. In the enzymatic assay, the test samples (100 µL) were mixed with Tris buffer (400 µL), substrate solution (350 µL), and pancreatic lipase (150 µL). The mixture was then incubated at 37 °C for 2 h and centrifugated at 4000 g for 3 min. The absorbance of the reaction mixture was measured at 400 nm by using a SpectraMax M5 microplate reader. The percent inhibition of pancreatic lipase activity was calculated by Equation (1).

### 3.7. Total Phenolics Content and Total Flavonoid Content

#### 3.7.1. Determination of Total Phenolics Content

The total phenolics content (TPC) of the crude extract was determined by the Folin–Ciocalteu’s colorimetric as per the method previously described but with slight modifications [41]. Briefly, the extract (100 µg/mL) was dissolved in methanol with a concentration of 1.0 mg/mL. Subsequently, 0.6 mL of each sample solution was added into 0.5 mL of Folin–Ciocalteu′s phenol reagent with 0.4 mL of distilled water, and the mixture was allowed to react for 1 min. Na_2_CO_3_ (*m*/*v*: 20%; 1.5 mL) and distilled water (6 mL) were successively added into the mixture, which was then heated in water bath at 70 °C for 10 min. After the reaction, the reagent was cooled to room temperature. The absorbance of the reaction mixture was measured at 765 nm by using a SpectraMax M5 microplate reader. The TPC in the extract was expressed as µg gallic acid equivalent (GAE) per mg of extract (µg GAE/mg extract).

#### 3.7.2. Determination of the Total Flavonoid Content

The total flavonoid content (TFC) content was determined by a previously described method with some modifications [42]. In brief, the extract (100 µg/mL) was dissolved in methanol to make a solution of 1.0 mg/mL, then 0.3 mL of 5% NaNO_2_ (*m*/*v*) and 3.8 mL of 70% ethanol (*v*/*v*) were added to 1.2 mL of the prepared sample solution, and allowed to stand for 8 min. Subsequently, 0.3 mL of 10% Al(NO_3_)_3_ (*m*/*v*) was added to the mixture followed by the addition of 4.0 mL of 4% NaOH and 0.4 mL of 70% ethanol (*v*/*v*). The mixture was stored at room temperature for 12 min, of which the absorbance was measured at 510 nm by using a SpectraMax M5 microplate reader. The DPPH radical scavenging activity was expressed as µg rutin equivalent (RE) per mg of extract (µg RE/mg extract).

### 3.8. Antioxidant Activity Assays

#### 3.8.1. DPPH Free-Radical Scavenging Assay

The DPPH free radical scavenging activity was carried out by the method previously described [43]. Briefly, 2 mL of DPPH solution (0.1 mmol/L) was mixed with 0.5 mL of sample solutions or Trolox solution (100 μg/mL in methanol). Then, the mixture was incubated for 30 min in the dark at room temperature. Absorbance of the mixture was measured at 517 nm using a SpectraMax M5 microplate reader. DPPH radical scavenging activity of each extract was expressed as nmol Trolox equivalent (TE) per mg of extract (nmol TE/mg extract).

#### 3.8.2. ABTS Radical Scavenging Assay

The ABTS assay was performed according to the procedure described earlier [44]. Briefly, 7 mM ABTS was added to 2.5 mM phosphoric acid buffer solution and the mixture was allowed to stand in the dark at room temperature for 16 h to obtain the ABTS radical solution. The ABTS solution was diluted with methanol with an absorbance of 0.70 ± 0.02 at 734 nm. After adding the 25 μL extract (100 μg/mL in methanol) to 200 μL ABTS solution, the absorbance was measured after 6 min at 734 nm by using a SpectraMax M5 microplate reader. Results were expressed as nmol Trolox equivalent (TE) per mg of extract (nmol TE/mg extract).

#### 3.8.3. Ferric Reducing/Antioxidant Power (FRAP) Assay

The FRAP assay was conducted according to the procedure reported with slight modifications [19]. The FRAP reagent was freshly prepared with 300 mM acetate buffer (pH 3.6), 10 mM TPTZ (2,4,6-tripyridyl-s-triazine), and 20 mM FeCl_3_-6H_2_O (10:1:1, *v*/*v*)]. A 0.5 mL of each sample (100 μg/mL) was mixed with 4.5 mL freshly prepared FRAP reagent. Then, the reaction mixture was incubated at 37 °C in the dark for 10 min. The absorbance of the mixture was recorded at 593 nm against a blank that was prepared using distilled water. Trolox was used as a control. The FRAP values were expressed as nmol Trolox equivalent (TE) per mg of extract (nmol TE/mg extract).

### 3.9. Characterization and Quantification of Isolated Compounds in the Extracts and the Fractions

A Thermo Fisher Ultimate 3000 UHPLC system (Thermo Fisher Scientific, Bremen, Germany) equipped with a C18 column (2.1 × 100 mm, 1.9 µm, Agilent, USA) was used to analyze the main phenolic compounds in the CE, PM, and fractions. The chromatographic conditions were optimized to achieve a good simultaneous separation of the major compounds. Acidified water (0.1% formic acid, solvent A) and acetonitrile (0.1% formic acid, solvent B) were prepared and used as mobile phases. A gradient program was adapted as follows: 0–30 min, 10–60% B; 30–35 min, 100% B. The flow rate was 0.2 mL/min, the injection volume was 2 µL, and the column temperature was maintained at 35 °C. The HR-ESI-MS/MS data were analyzed on a Q-Exactive Orbitrap mass spectrometer (Thermo Fisher Scientific, Germany) as per our previously reported condition [45]. The specific parameters performed: resolution, 70,000; auxiliary gas flow, 8 L/min; sheath gas flow rate, 32 L/min; sweep gas, 4 L/min; S-lens RF level, 50%; spray voltage, 3.3 kV, capillary temperature, 320 °C; and heater temperature, 350 °C. The ions were scanned in negative mode with a mass range from *m/z* 50 to 1000. The corresponding standards were used to quantify all the compounds identified in various fractions under the same conditions. The quantitative analysis was carried out using external standard calibration curves based on peak areas.

### 3.10. Statistical Analysis

All experiments were carried out at least in triplicate. The data were also analyzed by one-way analysis of variance (one-way ANOVA). Tukey’s test was applied to determine the significance of differences (*p* < 0.05). Principal component analysis (PCA) used Origin 8.5 software (OriginLab, Northampton, MA, USA). All other analyses were performed using SPSS Statistics 17.0 software.

## 4. Conclusions

The antioxidants and enzyme inhibitory effects of *V. dunalianum* were first carried out in the present work. PM, Fr. B, and Fr. C with a high content of TPC and TFC were the most bioactive fractions. The phenolic and flavonoids were positively related to the antioxidant, and the α-glucosidase and pancreatic lipase inhibitory activities. In addition, 10 pure compounds were identified by a bioactivity-guide isolation strategy. Arbutin derivatives were the main chemical constituents in *V. dunalianum* buds. 6′-*O*-Caffeoylarbutin (**7**) was the most abundant compound, and might also be the major biological ingredient. However, there is also a need to investigate the special mechanism of antioxidant or pancreatic lipase inhibitory effect of bioactive agents in further experiments in vitro and/or in vivo. In addition, more phytochemicals with lower content or significant biological activities should be found, and the synergistic effect between different compounds should be determined. Taken together, the extract or isolates from *V. dunalianum* could be used as a potential natural antioxidant and antidiabetic agents. These findings could provide a material basis for the traditional applications of *V. dunalianum* in human benefits.

## Figures and Tables

**Figure 1 molecules-26-02075-f001:**
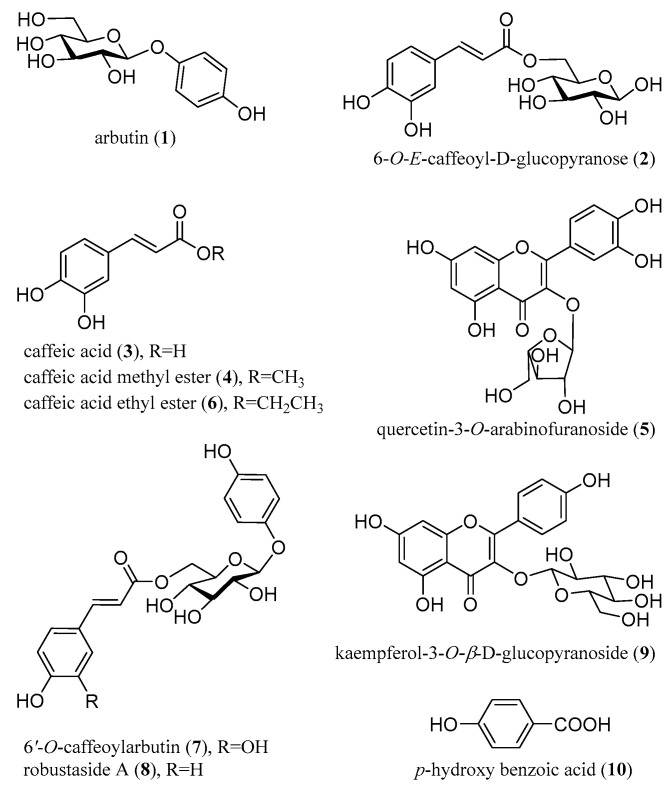
Structures of the isolated compounds (**1**–**10**).

**Figure 2 molecules-26-02075-f002:**
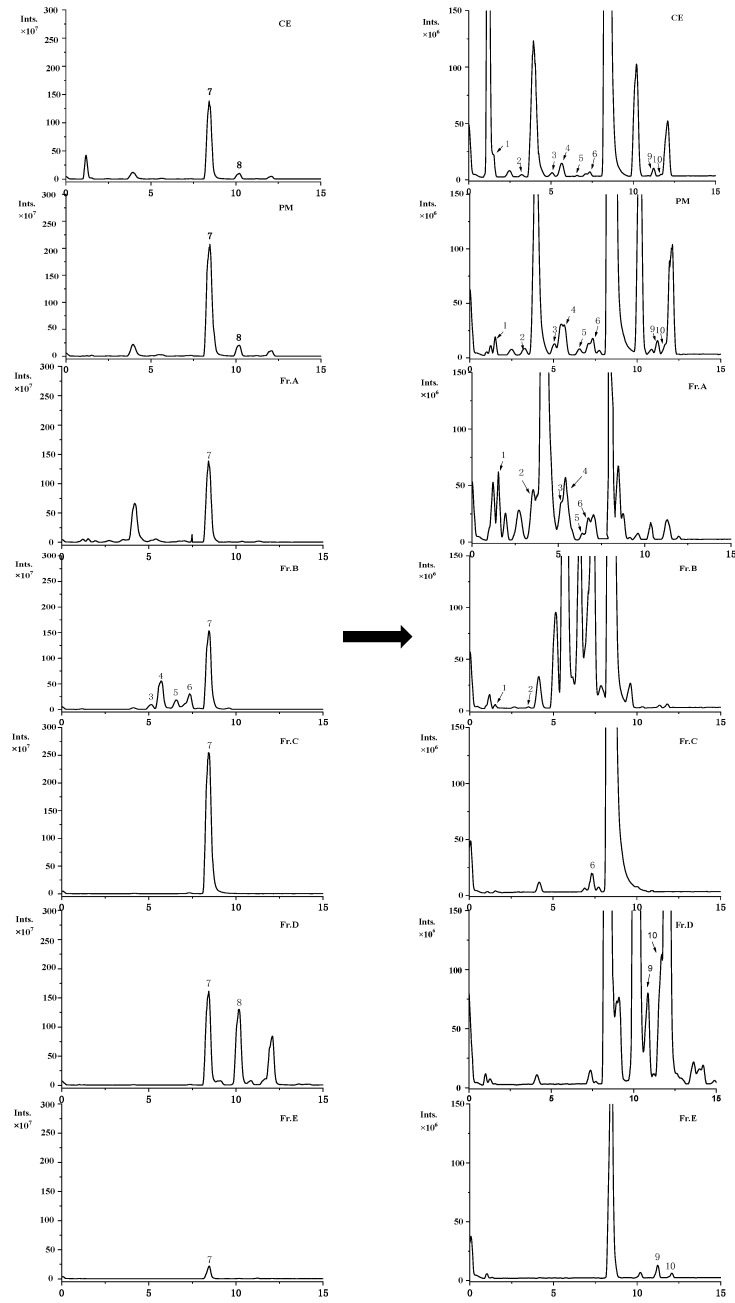
Negative ion chemical profiles of CE, PM, and Fr. A–E from *V. dunalianum* buds. CE: crude extract, PM: purified extract by macroporous resin D101, and Fr. A–E are the five fractions.

**Figure 3 molecules-26-02075-f003:**
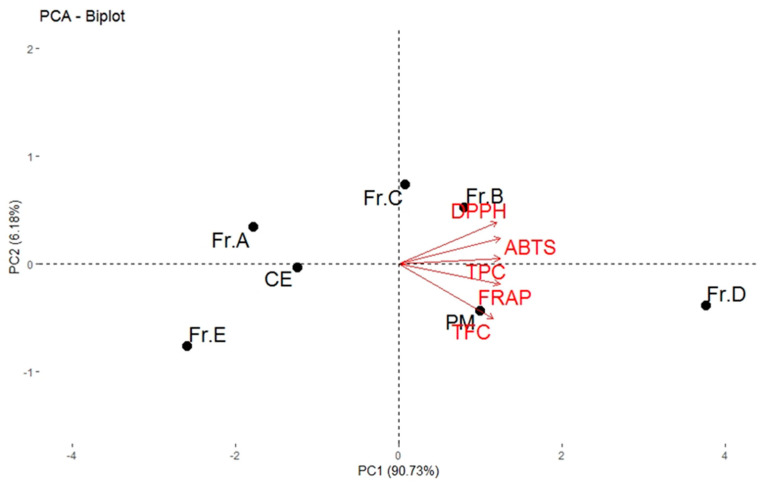
Principal component analysis (PCA) biplot. Effect of CE, PM, and fractions from *V. dunalianum* buds on total phenolics, flavonoids, α-glucosidase, pancreatic lipase inhibitory effect, and antioxidant activity. Bar values with different letters are significantly different (*p* < 0.05). CE: crude extract, PM: purified extract, Fr. A–E mean five fractions.

**Figure 4 molecules-26-02075-f004:**
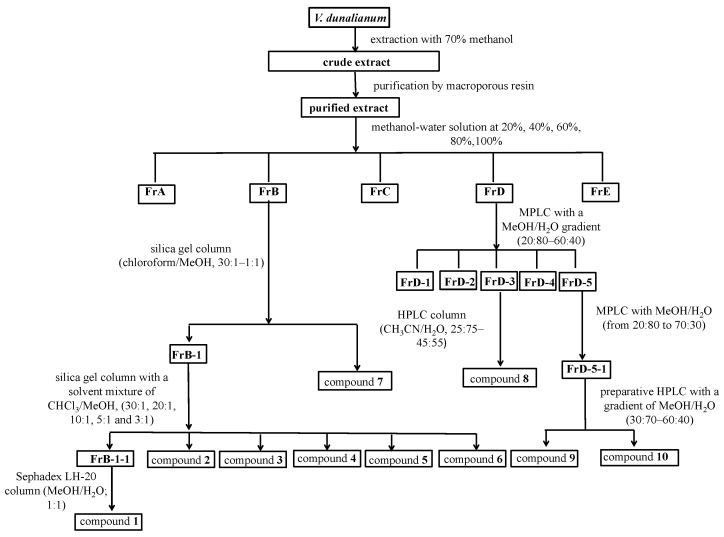
Extraction and isolation procedure of compounds from the 70% methanol extract of *V. dunalianum* buds.

**Table 1 molecules-26-02075-t001:** Chemical compositions of *V. dunalianum* buds by using UHPLC-ESI-HRMS/MS in negative mode.

Compounds	Retention Time (min)	[M−H]^−^ (*m/z*)	Error (ppm)	Molecular Formula	MS/MS Fragment Ions
**1**	1.49	271.0821	2.044	C_12_H_16_O_7_	86.0621, 100.0832
**2**	3.18	341.0877	3.024	C_15_H_18_O_9_	135.0421, 161.0263, 179.0374, 221.0415
**3**	5.08	179.0341	1.144	C_9_H_8_O_4_	89.0441, 134.0428, 135.0407
**4**	5.65	193.0491	1.452	C_10_H_10_O_4_	91.0261, 109.0306, 137.0206
**5**	6.58	433.0847	2.868	C_20_H_18_O_11_	300.9901, 257.0263
**6**	7.37	207.0611	1.343	C_11_H_12_O_4_	192.0414, 17.0274, 149.0214
**7**	8.43	433.1137	1.886	C_21_H_22_O_10_	161.0286, 179.0353
**8**	10.20	417.1224	2.712	C_21_H_22_O_9_	145.0352, 163.0403
**9**	11.25	447.0996	2.453	C_21_H_20_O_11_	145.0331, 160.0262, 175.0441, 193.0508
**10**	11.69	137.0233	0.296	C_7_H_6_O_3_	52.0342, 65.0442, 93.0317, 94.0426

**Table 2 molecules-26-02075-t002:** Quantitative results of the identified compounds in different extracts (CE, PM, and Fr. A–E) by UHPLC-ESI-HRMS/MS.

Compounds	CE	PM	Fr. A	Fr. B	Fr. C	Fr. D	Fr. E
**1**	33,415.57 ± 536.23^b^	34,435.62 ± 398.11 ^b^	48,580.19 ± 336.91 ^c^	14,333.86 ± 252.76 ^a^	-	-	-
**2**	324.31 ± 19.66 ^a^	1576.45 ± 56.39 ^b^	13,836.34 ± 314.39 ^d^	9842.77 ± 172.89 ^c^	-	-	-
**3**	9.21 ± 1.02 ^a^	90.14 ± 5.84 ^b^	7609.17 ± 239.03 ^c^	65,943.33 ± 802.34 ^d^	-	-	-
**4**	621.85 ± 89.11 ^a^	3065.83 ± 153.03 ^b^	3964.94 ± 148.17 ^c^	37,311.81 ± 302.41 ^d^	-	-	-
**5**	203.63 ± 12.96 ^a^	1361.53 ± 63.18 ^b^	402.12 ± 73.96 ^b^	36,193.44 ± 428.09 ^c^	-	-	-
**6**	160.22 ± 9.19 ^a^	1022.45 ± 43.80 ^b^	1330.61 ± 44.71 ^c^	21,689.76 ± 401.55 ^e^	737.82 ± 38.09^d^	-	-
**7**	313,974.28 ± 4479.73 ^d^	764,476.99 ± 6844.89 ^e^	65,757.63 ± 512.92 ^b^	406,201.66 ± 5895.22 ^c^	983,345.41 ± 8992.17 ^f^	415,741.09 ± 3988.21 ^c^	43,426.759 ± 593.76 ^a^
**8**	20,732.55 ± 419.74 ^a^	62,537.46 ± 848.04 ^b^	-	-	-	298,710.50 ± 3023.29 ^c^	-
**9**	1012.62 ± 59.87 ^a^	1876.33 ± 166.09 ^c^	-	-	-	17,644.37 ± 227.61 ^d^	1669.69 ± 55.69 ^b^
**10**	6959.61 ± 237.93 ^a^	78,904.11 ± 1037.96 ^c^	-	-	-	64,451.18 ± 556.14 ^d^	20427.45 ± 559.21^b^

bValues are expressed as the mean ± SD (*n* = 3). Different lower case superscript letters in the same row mean different significance (*p* < 0.05). The results are expressed as µg/g of the extract. CE (747 g): crude extract, PM (531 g): purified extract, Fr. A–E (70 g, 56 g, 65 g, 24 g, and 101 g) are the five fractions from the PM.

**Table 3 molecules-26-02075-t003:** Total phenolics, total flavonoids, enzyme inhibitory, and antioxidant activities of different extracts.

Sample	Amount ^1^	TPC ^2^	TFC ^3^	Enzyme Inhibition IC_50_ (µg/mL)		Antioxidant Activity		
				*ɑ*-glucosidase	Pancreatic Lipase	DPPH ^4^	FRAP ^5^	ABTS ^6^
CE	186.75	12.22 ± 0.25 ^c^	5.14 ± 0.32 ^c^	198.56 ± 9.34 ^i^	214.32 ± 17.53 ^d^	0.26 ± 0.01 ^b^	0.71 ± 0.04 ^c^	8.25 ± 0.12 ^f^
PM	132.75	36.16 ± 0.19 ^e^	14.39 ± 0.19 ^f^	109.75 ± 1.34 ^h^	165.43 ± 10.35 ^c^	0.40 ± 0.02 ^e^	0.91 ± 0.04 ^g^	12.87 ± 0.13 ^h^
Fr. A	17.5	10.15 ± 0.22 ^b^	4.12 ± 0.2 ^b^	202.36 ± 4.02 ^i^	235.21 ± 5.86 ^e^	0.32 ± 0.02 ^c^	0.37 ± 0.01 ^a^	5.52 ± 0.25 ^d^
Fr. B	14.0	30.93 ± 0.23 ^d^	9.42 ± 0.19 ^d^	95.21 ± 2.21 ^g^	142.53 ± 11.45 ^bc^	0.48 ± 0.03 ^ef^	0.87 ± 0.02 ^f^	14.42 ± 0.41 ^i^
Fr. C	16.25	36.11 ± 0.12 ^e^	5.64 ± 0.29 ^e^	101.76 ± 6.61 ^g,h^	155.52 ± 11.86 ^c^	0.42 ± 0.02 ^e^	0.69 ± 0.01 ^c^	11.45 ± 0.33 ^g^
Fr. D	6.0	56.15 ± 0.39 ^f^	20.31 ± 0.18 ^g^	74.55 ± 3.85 ^f^	128.76 ± 13.85 ^b^	0.62 ± 0.01 ^g^	1.58 ± 0.02 ^h^	19.25 ± 0.23 ^k^
Fr. E	25.25	8.46 ± 0.24 ^a^	5.74 ±0.04 ^a^	303.03 ± 7.27 ^j^	589.65 ± 10.98 ^j^	0.12 ± 0.01 ^a^	0.36 ± 0.01 ^a^	1.32 ± 0.14 ^a^
**1**		-	-	61.36 ± 1.85 ^e^	352.31 ± 18.92 ^i^	0.45 ± 0.02 ^e^	0.76 ± 0.01 ^d^	8.38 ± 3.74 ^efg^
**2**		-	-	42.43 ± 2.87 ^b^	103.86 ± 3.01 ^a^	1.05 ± 0.11 ^h^	0.84 ± 0.01 ^e,f^	15.93 ± 1.65 ^ij^
**3**		-	-	70.68 ± 0.73 ^f^	235.22 ± 14.04 ^e^	0.39 ± 0.05 ^de^	0.85 ± 0.01 ^ef^	3.86 ± 0.48 ^b^
**4**		-	-	49.61 ± 2.20 ^c^	267.34 ± 23.65 ^f^	0.37 ± 0.01^d^	0.82 ± 0.01 ^e^	3.10 ± 1.07 ^b^
**5**		-	-	55.25 ± 1.36 ^d^	329.56 ± 4.56 ^h^	0.81 ± 0.17 ^h^	-	-
**6**		-	-	53.31 ± 1.15 ^d^	257.55 ± 15.27 ^ef^	0.42 ± 0.03 ^e^	0.76 ± 0.01 ^d^	4.16 ± 0.25 ^b^
**7**		-	-	38.38 ± 1.84 ^a^	97.56 ± 7.53 ^a^	1.21 ± 0.43 ^c^	0.87 ± 0.01 ^f^	16.96 ± 0.95 ^j^
**8**		-	-	50.27 ± 2.43 ^c^	310.02 ± 12.82 ^g^	0.41 ± 0.06 ^e^	0.49 ± 0.01 ^b^	4.71 ± 0.23 ^c^
**9**		-	-	50.41 ± 1.01 ^c^	348.65 ± 18.95 ^i^	0.35 ± 0.01 ^cd^	-	6.38 ± 0.34 ^e^
**10**		-	-	63.34 ± 2.30 ^e^	306.21 ± 21.26 ^g^	1.12 ± 0.23 ^h^	0.87 ± 0.01 ^f^	-

^1^ Amount: expressed as mg/g dry weight of sample. ^2^ TPC: expressed as µg GAE equivalent/mg extract. ^3^ TFC: expressed as µg RE equivalent/mg extract. ^4^ DPPH: Expressed as nmol Trolox equivalent/mg extract. ^5^ FRAP: Expressed as nmol Trolox equivalent/mg extract. ^6^ ABTS: Expressed as nmol Trolox equivalent/mg extract. Data are from three replicates with mean ± SD; different superscript letter/s in the same column were significantly (*p* < 0.05) different.

## Data Availability

Data contained within the article or supplementary material.

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
