# Peer review of "Bioactivity-Guided Isolation of Phytochemicals from Vaccinium dunalianum Wight and Their Antioxidant and Enzyme Inhibitory Activities"

_molecules, 2021, doi:10.3390/molecules26072075_

Round 1

Reviewer 1 Report

This manuscript investigates the isolation of the selected phytochemicals from V.dunalianum wight, followed by the in vitro antioxidant and the target enzyme inhibitory properties determination. Although the research topic is interesting, there are specific comments addressed to the authors, which should be clarified prior to the further steps of publishing:

Abstract:

  • Line 25: “6′-O-caffeoylarbutin (7) showed significant inhibitory effect..”. This statement is misleading, since the manuscript does not provide data on the activity of any other known inhibitor. Therefore, the significance of inhibition without such comparison cannot be assessed easily.
  • Lines 27-28. “The results revealed that the species V. dunalianum or its extract may be useful in the development of functional food”. This statement is rather misleading given the limited applicability of methanolic extracts in food industry.

Keywords:

  • Keyword “Functional food” should be removed from this list, since none of the described experiments deals with the actual functional food development.

Introduction:

  • General comment: the introduction is too generic (especially, lines 32-55) and lacks more thorough discussion on the novelty of this research.
  • Line 61: “invest igations” should be changed into “investigations”;
  • Authors should provide clear justification, why the pancreatic lipase inhibitory activities were chosen to test in line with the α-glucosidase inhibiting capacity, and how these properties could be connected with the antioxidant capacity of samples?
  • Lines 68-69. “However, a relationship between chemical compounds and health beneficial activities is limited.” Authors should rephrase the sentence.

Results and Discussion:

  • Tables 1 and 3. Authors should clearly indicate the yields of crude extract, purified material, fractions and individual compounds. Also, it is not clear how the final values in these tables are expressed, it is per mass unit of crude extract (“starting point”) or per mass unit of each sample/fraction tested? If only the latter case is taken into consideration, the actual comparison between different samples might lead to the false observations. Therefore, authors should provide data on both: values per mass unit of sample, further recalculated to values per mass unit of crude extract.
  • Table 3. Authors provide data indicating the misleading mass balance in the case of Fraction D. When summed up, all reported compounds constitute 1376609.7 µg/g extract, or ~1.377 g/g extract. Also, unexplained fluctuation in the recovery of compound 7 is reported shifting from Fraction A to Fraction C. For example, similar amount of this compound could be obtained either with 20% or 100% MeOH/H2O solvent system. Authors should clearly address this issue in the manuscript.
  • Quality of Fig.2 should be improved.
  • Authors should explain the rationale behind the selection of UAE as the extraction technique and 70/30% MeOH/H2O as solvent system to prepare crude extract. What is the relevance of hydromethanolic extraction taking into consideration the possibilities to utilize these fractions for functional food development? Thus, the question arises why other more green solvent systems (e.g., ethanol-based ones) were not tested in order to meet the requirements of the modern food industry and, potentially, to maximize the recovery of target bioactive compounds.
  • Lines 113-114: “Thus, macroporous resin chromatography effectively enriched the bioactive compounds which was coherent with our previous results [19].” Authors should be more specific, indicating the target bioactive properties.
  • General comment: the discussion of the obtained results should be improved, providing more thorough explanations towards the obtained results, based both on the experimental observations and previous investigations of other researchers.

Materials and Methods:

  • Lines 277-280. Authors should indicate sample collection period, drying and storage conditions.
  • Line 283. Authors should indicate the mesh size of the ground sample.
  • Lines 392, 431, 442, 542, 466-467. Authors should indicate in all assays the concentration range of samples.
  • Line 452. Authors should indicate the amounts of sample and ABTS reagent used for the analysis.
  • Authors should describe how the IC50 values were calculated in α-glucosidase and pancreatic lipase inhibition assays.
  • Lines 482-485. Authors should be more precise in describing statistical data handling and do not refer to a Duncan's test as to a “Duncan procedure”. Also, authors should indicate how the correlation coefficients were calculated.
  • Lines 487-501. Information, provided in the template, should be removed from the manuscript.

Author Response

Detailed list of all responses to the editor

Thank you very much for your valuable comments. We have revised our manuscript carefully according to your comments and suggestions. All changes in the manuscript have been highlighted in red. All the co-authors have read and approved the revised manuscript.

Editor and Reviewer 1 Comments:

This manuscript investigates the isolation of the selected phytochemicals from V.dunalianum wight, followed by the in vitro antioxidant and the target enzyme inhibitory properties determination. Although the research topic is interesting, there are specific comments addressed to the authors, which should be clarified prior to the further steps of publishing:

Abstract:

  • Line 25: “6′-O-caffeoylarbutin (7) showed significant inhibitory effect.” This statement is misleading, since the manuscript does not provide data on the activity of any other known inhibitor. Therefore, the significance of inhibition without such comparison cannot be assessed easily.

Answer: Thank you for your valuable suggestion. We have changed the sentence “6′-O-caffeoylarbutin (7) showed significant inhibitory effect..” to “6-O-caffeoylarbutin (7) showed significant inhibitory effects on α-glucosidase and pancreatic lipase enzymes with values of 38.38 ± 1.84 and 97.56 ± 7.53 µg/mL, and had the highest antioxidant capacity than other compounds.”.

  • Lines 27-28. “The results revealed that the species  dunalianum or its extract may be useful in the development of functional food”. This statement is rather misleading given the limited applicability of methanolic extracts in food industry.

Answer: Thanks for your suggestion. The methanol is not potential solvent in the food industry due to its potent toxicity. We had throughout deleted the applicability of methanolic extracts in food industry and made a relevant revision in the manuscript.

Keywords:

  • Keyword “Functional food” should be removed from this list, since none of the described experiments deals with the actual functional food development.

Answer: We have deleted the keyword. Thank you!

Introduction:

General comment: the introduction is too generic (especially, lines 32-55) and lacks more thorough discussion on the novelty of this research.

Answer: We had made a revision. Thank you!

  • Line 61:“invest igations” should be changed into “investigations”;

Answer: We have modified it. Thank you!

  • Authors should provide clear justification, why the pancreatic lipase inhibitory activities were chosen to test in line with the α-glucosidase inhibiting capacity, and how these properties could be connected with the antioxidant capacity of samples?

Answer: Thanks for your suggestion. Vaccinium dunalianum Wight were used as a traditional Chinese medicine in the treatment of diabetes and hyperlipemia, so in order to further research its folk use, we investigated the inhibitory effect of the extract on pancreatic lipase and α-glucosidase. The antidiabetic was also determined on two key enzymes (pancreatic lipase and α-glucosidase) in human. In addition, oxidative stress is also related to a human disease: diabetes mellitus. In human, antioxidants and antioxidant enzymes together form an antioxidant system to reduce the overexpression of active oxygen in the body and achieve effective control of oxidative stress. Therefore, we have chosen these three evaluation methods.

  • Lines 68-69. “However, a relationship between chemical compounds and health beneficial activities is limited.” Authors should rephrase the sentence.

Answer: We changed it to “However, these studies are mainly focused on the phytochemicals with novel structures of arbutin derivatives. The anti-diabetic and hypolipidemic activity evaluation on the extract and pure compound from V. dunalianum is limited”. Thank you!

Results and Discussion:

Tables 1 and 3. Authors should clearly indicate the yields of crude extract, purified material, fractions and individual compounds. Also, it is not clear how the final values in these tables are expressed, it is per mass unit of crude extract (“starting point”) or per mass unit of each sample/fraction tested? If only the latter case is taken into consideration, the actual comparison between different samples might lead to the false observations. Therefore, authors should provide data on both: values per mass unit of sample, further recalculated to values per mass unit of crude extract.

Answer: Thanks for your suggestion. We have added extraction amount of crude extract purified material, fractions in the manuscript and table 1, which were expressed as milligram per gram dried weight of sample. For the individual compounds, the amount of each compound after a series of isolation procedures is not accurate. Furthermore, the contents of individual compounds were tested by UHPLC-ESI-MS/MS method, and the results were described in Table 2.

  • Table 3. Authors provide data indicating the misleading mass balance in the case of Fraction D. When summed up, all reported compounds constitute 1376609.7 µg/g extract, or 1.377 g/g extract. Also, unexplained fluctuation in the recovery of compound 7 is reported shifting from Fraction A to Fraction C. For example, similar amount of this compound could be obtained either with 20% or 100% MeOH/H2O solvent system. Authors should clearly address this issue in the manuscript.

Answer: Thanks for your comment. This is our mistake in the content of compound 10. We had rechecked it again and revised it in Table 2. The found of compound 7 in all of five fractions may be related to the adsorption capacity of macroporous resins, desorption capacity of the eluting methanol concentration, as well as the solubility of compossible. In addition, 5 column volumes of per gradient of methanol-water system may be insufficient for the separation of compound 7. In the future, we will investigate the adsorption and desorption properties of different macroporous resins for the enrichment and purification of compound 7.

  • Quality of 2 should be improved.

Answer: we have improved quality of Fig.2. Thank you!

  • Authors should explain the rationale behind the selection of UAE as the extraction technique and 70/30% MeOH/H2O as solvent system to prepare crude extract. What is the relevance of hydromethanolic extraction taking into consideration the possibilities to utilize these fractions for functional food development? Thus, the question arises why other more green solvent systems (e.g., ethanol-based ones) were not tested in order to meet the requirements of the modern food industry and, potentially, to maximize the recovery of target bioactive compounds.

Answer: In our preliminary experiment, methanol-aqueous solvent had a higher extraction yield than the water, ethanol-aqueous solvent, and acetone-aqueous solvent. Thus, the aim of this paper was to isolation of the bioactive compounds. So that, we do not consider the safety of solvent systems. Thanks very much for your suggestion about the green and safety solvents selected for the development of foods in the food industry.

  • Lines 113-114: “Thus, macroporous resin chromatography effectively enriched the bioactive compounds which was coherent with our previous results [19].” Authors should be more specific, indicating the target bioactive properties.

Answer: we have revised the manuscript and gave the specific data before and after macroporous resin chromatography. Thanks.

General comment: the discussion of the obtained results should be improved, providing more thorough explanations towards the obtained results, based both on the experimental observations and previous investigations of other researchers.

Answer: We have improved the results and discussion. Thanks.

Materials and Methods:

  • Lines 277-280. Authors should indicate sample collection period, drying and storage conditions.

Answer: Thanks for your valuable suggestion. We have indicated these conditions in the manuscript.

  • Line 283. Authors should indicate the mesh size of the ground sample.

Answer: We have added the mesh size of the ground sample “ the dried sample (4.0 kg) was powdered and passed by 100-mesh sieve …” in paper.

  • Lines 392, 431, 442, 542, 466-467. Authors should indicate in all assays the concentration range of samples.

Answer: We have added those concentration range. Thank you!

  • Line 452. Authors should indicate the amounts of sample and ABTS reagent used for the analysis. Authors should describe how the IC50 values were calculated in α-glucosidase and pancreatic lipase inhibition assays.

Answer: Thanks for your valuable suggestion. We have increased the amounts of sample and ABTS used in the manuscript.

The inhibitory activities of α-glucosidase and pancreatic lipase were measured by setting the concentration gradient of the sample, and then the IC50 values were calculated using data analysis in SPSS Statistics 17.0 software.

  • Lines 482-485. Authors should be more precise in describing statistical data handling and do not refer to a Duncan's test as to a “Duncan procedure”. Also, authors should indicate how the correlation coefficients were calculated.

Answer: We have revised the part of “Statistical analysis” and provided the data processing method.

  • Lines 487-501. Information, provided in the template, should be removed from the manuscript.

Answer: Sorry, we have modified those errors.

Reviewer 2 Report

In this paper, antioxidant and enzyme inhibitory activities of V. dunalianum extract as well as its fractions were investigated. The topic of the paper is interesting. However, some revisions are required to improve the quality of the paper. Some comments are listed below:

  • In several points in the manuscript antioxidant and enzyme inhibitory activities of V. dunalianum extract as whole have been mentioned, and in the conclusion it was concluded that V. dunalianum could be a potential source for the development of functional beverages and food supplements in food industry. So, it is not very clear why the authors fractionated the extracts, I think it should be mentioned and described better to clarify.
  • In the last paragraph of the introduction “the bioactivity-guided isolation from antioxidant fractions was performed to uncover the main bioactive compounds” please revise and clarify this sentence. It is not neccessary to perform isolation-fractionation to find out the bioactive compounds.
  • Under “4.10. Statistical analysis” section the paragraph starting with “The Materials and Methods should be described with sufficient…” and the following 2 paragraghs should be removed. I think it is mistakenly left.
  • In the abstract and conclusion sections, it would be better to describe these fractions instead of just writing Fr B and Fr D, etc. In this form it is not very clear.
  • Sometimes it was written as “Fr.” Sometimes as “Fr” to abbreviate “fraction”, please make it uniform throughout the whole text.
  • The discussion needs to be improved in general. In most cases, only the results have been provided without further discussion. For example, the trends for total phenolics, total flavonoids or total antioxidant activity of different fractions are not the same. Please discuss this in more detail in the manuscript. Also, in Table 1 antioxidant activity of some compounds was not detected by some methods even though high values were obtained with the other techniques. Please discuss these results.
  • Please check the statistical lettering in Table 1, there are some mistakes.

Author Response

Editor and Reviewer 2 Comments:

  • In this paper, antioxidant and enzyme inhibitory activities of dunalianum extract as well as its fractions were investigated. The topic of the paper is interesting. However, some revisions are required to improve the quality of the paper. Some comments are listed below:

In several points in the manuscript antioxidant and enzyme inhibitory activities of V. dunalianum extract as whole have been mentioned, and in the conclusion it was concluded that V. dunalianum could be a potential source for the development of functional beverages and food supplements in food industry. So, it is not very clear why the authors fractionated the extracts, I think it should be mentioned and described better to clarify.

Answer: The extract of V. dunalianum is a complex mixture system with various compounds. In order to find out the main bioactive compounds relating to its traditional use, a bioactivity-guided isolation was performed. We have revised the results and discussion in the manuscript.

In the last paragraph of the introduction “the bioactivity-guided isolation from antioxidant fractions was performed to uncover the main bioactive compounds” please revise and clarify this sentence. It is not neccessary to perform isolation-fractionation to find out the bioactive compounds.

Answer: Thanks for your suggestion! We have re-wrote the last paragraph of the introduction.

Under “4.10. Statistical analysis” section the paragraph starting with “The Materials and Methods should be described with sufficient…” and the following 2 paragraghs should be removed. I think it is mistakenly left.

Answer: We have corrected the mistakes. Thank you!

In the abstract and conclusion sections, it would be better to describe these fractions instead of just writing Fr B and Fr D, etc. In this form it is not very clear.

Answer: Thanks for your advice! We have modified it.

Sometimes it was written as “Fr.” Sometimes as “Fr” to abbreviate “fraction”, please make it uniform throughout the whole text.

Answer: Thanks for your advice! We have revised it throughout the manuscript.

The discussion needs to be improved in general. In most cases, only the results have been provided without further discussion. For example, the trends for total phenolics, total flavonoids or total antioxidant activity of different fractions are not the same. Please discuss this in more detail in the manuscript.

Answer: We have improved the discussion, as follows:

Moreover the isolated compounds were found to be more potent on enzyme of α-glycosidase than pancreatic lipase. Especially, 6′-O-caffeoylarbutin (7) has the best inhibitory activity on α-glycosidase than other compounds. Fr C possesses the maximum content of compound 7, but it is not the most bioactivity fraction on the inhibition against α-glycosidase and pancreatic lipase and antioxidant activity. Both Fr B and Fr D have the better activating capacity than Fr C, suggesting that some multiple-compound synergy effects in combination of 7 with other compounds. In addition, caffeic acid methyl ester (4) and caffeic acid ethyl ester (6) have been reported good antidiabetic effects for the activation of AMPK, and they have good ɑ-glucosidase inhibitory activities responsible for traditional practice in the treatment of diabetes[1,2].

References:

[1] Peyrat-Maillard, M.N.; Cuvelier, M.E.; Berset, C. Antioxidant activity of phenolic compounds in 2,2′-azobis (2-amidinopropane) dihydrochloride (AAPH)-induced oxidation: Synergistic and antagonistic effects. J. Am. Oil Chem. Soc., 2003, 80, 1007-1012.

[2] Eid, H.M.; Thong, F.; Nachar, A.; Haddad, P.S. Caffeic acid methyl and ethyl esters exert potential antidiabetic effects on glucose and lipid metabolism in cultured murine insulin-sensitive cells through mechanisms implicating activation of AMPK. Pharm. Biol. (Abingdon, U. K.), 2017, 55, 2026-2034.

Also, in Table 1 antioxidant activity of some compounds was not detected by some methods even though high values were obtained with the other techniques. Please discuss these results.

Answer: Three different methods were used for antioxidant evaluation, because different antioxidant methods had different mechanisms, which might be related to the type compounds, the numbers, the locations of hydroxy groups and glycosylation position [1].

References:

[1] Shang, A.; Liu H.; Luo M.; Xia Y.; Yang X.; Li H.; Sun Q.; Geng F.; Li H,; Gan R. Sweet tea (Lithocarpus polystachyus rehd.) as a new natural source of bioactive dihydrochalcones with multiple health benefits[J]. Critical Reviews in Food Science and Nutrition. 2020, (5), 1-8.

Please check the statistical lettering in Table 1, there are some mistakes.

Answer: Thanks for suggestions. We have modified the mistakes.

Reviewer 3 Report

Authors presented a well written MS on Vaccinium dunalianum. I have some minor comments as follows:

Add more information on Vaccinium dunalianum. Is it endemic to china?  other countries where it is used? How it is used? Why buds were used?

Also in abstract, add more values in the sentence "Bioactivity assays revealed that fractions B and D 23 have better antioxidant and enzyme inhibitory effects." Also add results of the PCA? What authors found, how it relates to the biological activity. Also, have authors been able to justify it traditional used. Please address this in the MS. "The results revealed that the species V. dunalianum or its extract may be useful in the development 27 of functional food." which functional food, geared towards which pathologies? Please ad more information. 

The relevance of inhibitory activities against :α-glucosidase and pancreatic lipase" should be elaborated in the discussion section. 

Author Response

Detailed list of all responses to the editor

Thank you very much for your valuable comments. We have revised our manuscript carefully according to your comments and suggestions. All changes in the manuscript have been highlighted in red. All the co-authors have read and approved the revised manuscript.

Editor and Reviewer 3 Comments:

Authors presented a well written MS on Vaccinium dunalianum. I have some minor comments as follows:

Add more information on Vaccinium dunalianum. Is it endemic to china?  other countries where it is used? How it is used? Why buds were used?

Answer: Thanks for your valuable suggestion. We have supplemented and revised the manuscript.

Answer: Vaccinium dunalianum Wight is an evergreen perennial shrub distributed in the Southwest of China, Myanmar and Vietnam. In Yunnan, V. dunalianum (Quezui tea) is an important herbal tea, and limited in the chemical constituents and the human health potentials. The local people of its growing area in China use the leaf of this plant as a folk medicine for the treatment of articular rheumatism, and also substitute its dried buds for tea to prepare a traditional folk beverage. [1, 2] In some minority areas, herbal plants are usually processed as a substitute of tea and are used as a traditional remedy for alleviating hypertension, diabetes, atherosclerosis, and maintaining healthy kidneys as well as relieving coughs. In other countries, it is well known that the Vaccinium species are rich sources of naturally compounds, such as flavonoids, anthocyanins, proanthocyanidins and other phenolic compounds.

References:

[1]   M. Zeng, Yunnan traditional Chinese medicine resource directory [M], Science Press, Beijing. 1993, p. 388.

[2]   Zhao, P.; Tanaka, T.; Hirabayashi, K.; Zhang, Y.J.; Yang, C.R.; Kouno, I. Caffeoyl arbutin and related compounds from the buds of Vaccinium dunalianum. Phytochemistry. 2008, 69, 3087-3094.

Also in abstract, add more values in the sentence "Bioactivity assays revealed that fractions B and D have better antioxidant and enzyme inhibitory effects." Also add results of the PCA? What authors found, how it relates to the biological activity.

Answer: Thanks for your valuable suggestion. We have added the data of antioxidant and enzyme inhibitory effects, and the PCA results in the manuscript.

Also, have authors been able to justify it traditional used. Please address this in the MS.

Answer: Thanks for your valuable suggestion. We have revised it in the manuscript.

"The results revealed that the species V. dunalianum or its extract may be useful in the development of functional food." which functional food, geared towards which pathologies? Please ad more information.

Answer: We have modified this sentence.

The relevance of inhibitory activities against: α-glucosidase and pancreatic lipase" should be elaborated in the discussion section.

Answer: Thanks for your valuable suggestion. We have revised it in the manuscript.

In current work, among these ten compounds, 6′-O-caffeoylarbutin (7) has the best antioxidant activity in DPPH (1.21 ± 0.43 nmol TE/g extract), FRAP (0.87 ± 0.01 nmol TE/g extract) and ABTS (16.96 ± 0.95 nmol TE/g extract), followed by 6-O-trans-caffeoyl-D-glucopyranose (2). The results were similar to the results of phenolic substances in highbush blueberry and rabbiteye blueberry (Vaccinium ashei) [1,2]. The values of TPC and TFC were highly correlated with DPPH values (r = 0.9114 and 0.7484), and strongly with FRAP (r = 0.9076 and 0.9205) and ABTS (r = 0.9311 and 0.8058). These results revealed that the TPC or TFC contents are responsible for the antioxidant capacity, which were consistent with the results observed in previous study [3]. The content of robustaside A (8), kaempferol-3-O-β-D-glucopyranoside (9) and 4-hydroxy benzoic acid (10) highly correlated with the values of antioxidants (r = 0.8948, 0.8469 and 0.8691, respectively, p < 0.05). It reported that the antioxidant capacity of different fractions was positively associated with phenolics [4]. In addition, we found the content of 6′-O-caffeoylarbutin had moderate correlation with the values of α-glucosidase and pancreatic lipase inhibitory activities (r =-0.7104, -0.6919, respectively, p < 0.05). There are also studies showed that caffeic acid (3) [5], quercetin-3-O-arabinofuranoside (5) [6], and kaempferol-3-O-β-D-glucopyranoside (9) [7] has a positive effect on antioxidant acitivities.

References:

[1] Rossi, M.; Giussani, E.; Morelli, R.; Scalzo, R.L.; Nani, R.C.; Torreggiani, D. Effect of fruit blanching on phenolics and radical scavenging activity of highbush blueberry juice. Food Res. Int., 2003, 36, 999-1005.

[2] Su, M.S.; Chien, P.J. Antioxidant activity, anthocyanins, and phenolics of rabbiteye blueberry (Vaccinium ashei) fluid products as affected by fermentation. Food Chem., 2007, 104, 182-187.

[3] Wojdy, O, A.; Oszmiański, J.; Czemerys, R.; Antioxidant activity and phenolic compounds in 32 selected herbs. Food Chem., 2007, 105, 940-949.

[4] Deetae, P.; Parichanon, P.; Trakunleewatthana, P.; Chanseetis, C.; Lertsiri, S. Antioxidant and anti-glycation properties of Thai herbal teas in comparison with conventional teas. Food Chem., 2012, 133, 953-959.

[5] Silva, A.M.; Pinto, D.; Fernandes, I.; Goncalves Albuquerque, T.; Costa, H.S.; Freitas, V.; Rodrigues, F.; Oliveira, M.B.P.P. Infusions and decoctions of dehydrated fruits of Actinidia arguta and Actinidia deliciosa: Bioactivity, radical scavenging activity and effects on cells viability. Food Chem., 2019, 289, 625-634.

[6] Mucaji, P.; Nagy, M.; Sersen, F.; Svajdlenka, E.; Drozd, J.; Stujber, M.; Liptaj, T. Phenolic metabolites from leaves of Karwinskia humboldtiana. Chem. Listy., 2012, 106, 1143-1146.

[7] Mojarrab, M.; Delazar, A.; Moghadam, S.B.; Nazemiyeh, H.; Nahar, L.; Kumarasamy, Y.; Asnaashari, S.; Hadjiakhoondi, A.; Sarker, S.D. Armenin and Isoarmenin - Two Prenylated Coumarins from the Aerial Parts of Artemisia armeniaca. Chem. Biodiversity, 2011, 8, 2097-2103.

Round 2

Reviewer 1 Report

Authors have performed all listed corrections and manuscript can be considered for publishing.